# Resveratrol and Physical Activity: A Successful Combination for the Maintenance of Health and Wellbeing?

**DOI:** 10.3390/nu17050837

**Published:** 2025-02-28

**Authors:** Mario Ruggiero, Maria Letizia Motti, Rosaria Meccariello, Filomena Mazzeo

**Affiliations:** 1Department of Medical, Human Movement and Well-Being Sciences, University of Naples Parthenope, 80133 Naples, Italy; mario.ruggiero005@studenti.uniparthenope.it (M.R.); letizia.motti@uniparthenope.it (M.L.M.); rosaria.meccariello@uniparthenope.it (R.M.); 2Department of Economics, Law, Cybersecurity and Sports Sciences, University of Naples Parthenope, 80035 Nola, Italy

**Keywords:** resveratrol, physical activity, sirtuins, inflammation, oxidative stress, ROS

## Abstract

Physical exercise is an essential component of human health. In recent years, scientific research has focused on identifying natural compounds and formulating new supplements aimed at enhancing athletic performance, accelerating muscle recovery, and minimizing the damage caused by physical exertion. The use of antioxidants to counteract the formation of reactive oxygen species (ROS) following physical activity (PA) is already a widely adopted practice. Resveratrol (RES), a polyphenol belonging to the stilbene class, is well known for its potent antioxidant activity and anti-inflammatory effects primarily attributed to the activation of sirtuins. RES possesses multiple nutraceutical properties used for the prevention and treatment of inflammatory, cardiovascular, neoplastic, and infectious diseases, thus attracting attention to study its use in combination with physical exercise to promote well-being. Animal trials combining RES and PA have mainly reported improvements in muscle, energy, and cardiovascular functions. The data presented and discussed in this narrative review are from Pubmed, Scopus, and the Human Gene Database (search limited to 2011 to 2025 with the keywords RES, sirtuins, and physical activity altogether or in combination with each other). This review gathers several studies on RES focusing on its nutraceutical properties, epigenetic activities via sirtuins, and the potential benefits of combining RES with PA in maintaining health and well-being based on trials performed first in animals and later in humans. Human studies have been conducted on various populations, including active adults, sedentary individuals, patients with diseases, and elderly individuals. Some studies have confirmed the benefits of RES observed in animal experiments. However, in some cases, no substantial differences were found between RES supplementation and the control group. In conclusion, the benefits of RES on PA reported in the literature are still not fully evident, given the contrasting studies and the still limited number of trials, but both RES and PA are successful tools for the maintenance of health and wellbeing.

## 1. Introduction

In recent years, numerous studies have confirmed the benefits of physical exercise for human health. The World Health Organization (WHO) recommends engaging in 150 to 300 min of physical activity (PA) per week, or 30 to 60 min per day, five days a week [1,2]. A sedentary lifestyle is associated with a higher incidence of chronic diseases and premature mortality [3,4]. The positive effects of exercise encompass both physical aspects—such as improvements in the cardiovascular system, muscle and bone health, metabolism, and weight control [3]—and mental aspects—including enhanced cognitive functions [5], better sleep quality, and reduced anxiety and depression [6]. Vina et al. describe PA as so effective that it should be considered a therapeutic intervention comparable to a drug [7].

However, physical exertion accelerates metabolism, leading to the production and release of reactive oxygen species (ROS) from mitochondria [8]. This process may impair performance by reducing the muscle contractile capacity, inducing fatigue [9], causing cellular damage, compromising the immune system [10], and triggering inflammatory responses [11].

Moreover, a well-planned diet that includes a variety of fruits, vegetables, whole grains, lean proteins, and healthy fats typically provides all the necessary vitamins and minerals, even for those people with increased PA [12]. For most athletes and active individuals, a well-balanced diet is sufficient to meet their nutritional needs, including vitamins and minerals. Prioritizing whole foods and a varied diet is the best approach [13].

Intense or prolonged exercise increases free radical production, leading to oxidative stress. Therefore, antioxidant supplementation is a widely used strategy to protect athletes from oxidative stress caused by physical exertion and to enhance the body’s antioxidant defense system [14,15]. While antioxidant supplementation is a common practice among athletes, it is crucial to be aware of the mixed evidence and potential negative effects. A balanced diet rich in antioxidants from whole foods is generally the best approach. If athletes are considering antioxidant supplements, they should consult with a healthcare professional or a registered dietitian to determine if this is appropriate for their individual needs.

Nutrition and PA are inextricably linked to lifelong health. By adopting a healthy lifestyle that incorporates both a balanced diet and regular PA, individuals can significantly improve their overall well-being and reduce their risk of chronic diseases, leading to a longer and healthier life. Similarly, the relevance of the manipulation of lifestyle factors such as dietary interventions is recognized in clinical practice and may represent a successful therapeutic approach to maintaining and preserving health along the lifespan [16,17,18,19,20,21].

Athletes frequently use antioxidant supplements to mitigate exercise-induced oxidative stress, enhance recovery, and improve performance. Among the most common exogenous antioxidants available through nutrition and supplementation are polyphenols [22], vitamin C, vitamin E, glutathione, carotenoids, coenzyme Q10 [23], selenium [24], and curcumin [25].

Polyphenols are secondary metabolites found in plants [26] and are present in fruits, vegetables, cereals, and beverages [27,28,29]. They are characterized by phenolic rings and hydroxyl groups [30]. They are classified into four main groups: phenolic acids, lignans, stilbenes, and flavonoids [31,32,33]. The antioxidant activity of polyphenols depends on their ability to donate electrons, stabilizing free radicals due to the presence of hydroxyl groups [34]. Polyphenols also have metal-chelating properties, which can help mitigate ROS that damage the DNA [35]. Moreover, polyphenols inhibit certain oxidative enzymes, such as xanthine oxidase and nicotinamide adenine dinucleotide phosphate (NADPH) oxidase, and up-regulate the activity of other antioxidant enzymes like glutathione peroxidase (GPX), catalase (CAT), and superoxide dismutase (SOD) [11].

The bioavailability of polyphenols is influenced by the size of the molecule [36], and their beneficial effects can be influenced by light, oxygen, temperature, gastrointestinal enzymatic activity, gut microbiota microorganisms that can convert phenolic compounds into more readily absorbed metabolites, pH, and temperature [37,38]. In this respect, to improve their bioavailability, polyphenols have been formulated into simple emulsions [39,40,41], cyclodextrins [42,43,44], gels [45,46,47], nanoemulsions [48,49,50], and liposomes [51,52,53].

Polyphenols recommended in the literature to improve athletic performance include resveratrol (RES) [54], quercetin [55], grape extract compounds [56], and beetroot extracts [57,58,59].

This narrative review focuses on RES, one of the main polyphenols primarily found in grapes, red wine, cocoa, and blueberries, among others, with numerous nutraceutical properties, particularly antioxidant, anti-inflammatory, and anti-aging effects, as detailed in the next paragraph. This review gathers several studies on RES, focusing on its nutraceutical properties, epigenetic activities via sirtuins, and the potential benefits of combining RES with PA in maintaining health and well-being, based on trials performed first in animals and later in humans, thus filling a knowledge gap in the field. Even though RES shows promising potential to enhance the effects of physical exercise, the data presented here from both animal models and clinical studies indicate the need for further researchers in the field to assess the real benefits and potential risks associated with RES use as a supplement for professional athletes and individuals engaging in PA.

The data presented and discussed in this narrative review were obtained from Pubmed, Scopus, and the Human Gene Database (restricted to the years 2011 to 2025 with the keywords RES, sirtuins, and physical activity altogether or in combination with each other).

## 2. Resveratrol

RES (trans-3,4′,5-trihydroxystilbene) is a natural polyphenol belonging to the stilbene class [60]. In plants, it functions as a phytoalexin [61], providing antibiotic and antifungal protection [62]. Beyond its protective role in plants, RES is widely recognized for its nutraceutical properties, which are utilized in the prevention and management of various diseases [63].

Takaoka was the first to isolate RES from the roots of *Veratrum grandifolium* [64,65]. RES is found in various fruits, vegetables, and processed foods, including grapes, red wine, blueberries, soy [66], peanuts, plums, cocoa, walnuts, *Polygonum cuspidatum* tea, and various seed oils [61,67] (Figure 1).

Some researchers have hypothesized that RES consumption through red wine may contribute to the low incidence of heart diseases in southern France despite a diet rich in saturated fats—a phenomenon known as the French paradox [69,70,71].

From a chemical standpoint, RES exists in two isomeric forms: trans-RES, which is more stable and biologically active, and cis-RES. It is poorly soluble in water [72]. Approximately 70% of RES is absorbed through the gastrointestinal tract; however, its hepatic and intestinal metabolism renders it poorly bioavailable [73]. In the bloodstream, RES can exist in three different forms (glucuronide, sulfate, or unmetabolized), which can bind to albumin and lipoproteins, further reducing its effectiveness [74]. This limitation affects its therapeutic efficacy, prompting research into strategies to enhance its absorption.

### 2.1. Health Outcomes

RES possesses numerous nutraceutical properties, including antioxidant, anti-inflammatory, anti-aging, and antiviral activities [75,76,77]. These characteristics make it useful in the prevention and management of cardiovascular diseases [78], metabolic syndrome [79], cancer [75,76], and oxidative stress [80], as summarized in Figure 2.

#### 2.1.1. RES, Oxidative Stress, and Inflammation

Oxidative stress is an excess of pro-oxidant agents due to an imbalance between pro-oxidants and antioxidants. RES acts as a potent antioxidant, controlling several antioxidant enzymes and blocking free radicals-dependent DNA damage [81]. Depending on the distribution of the functional groups in its structure, it is able to differentially activate alternative antioxidant mechanisms (radical scavenging/metal ion chelation) [82,83]. RES protects cells from oxidative stress caused by hydrogen peroxide and, in fact, it is able to prevent cell death caused by UV irradiation, promoting cell survival. It can act directly as an antioxidant or, alternatively, activate the antioxidant mechanisms of cells [84,85].

RES has strong anti-inflammatory properties, as it acts by inhibiting the production of anti-inflammatory factors [86].

Many studies have shown that its anti-inflammatory activity is exerted through action on different pathways, including the arachidonic acid pathway [87], the NF-κb pathway [88], the MAPK pathway [89], and the AP-1 pathway [90].

RES exerts anti-inflammatory effects on different systems and cell types. It has been found to be effective in inhibiting the proliferation of splenic lymphocytes induced by mitogens, interleukin (IL-2), or alloantigens, and the development of antigen-specific cytotoxic T lymphocytes (CTLs). It acts by blocking the production of IFN-gamma and IL-2 by splenic lymphocytes and the production of TNF-α and IL-12 by peritoneal macrophages. This inhibitory action of RES occurs through the inhibition of NF-κB activation, a transcription factor that regulates the expression of numerous pro-inflammatory genes, including cytokines and chemokines, and also participates in inflammasome regulation [91,92].

Also, RES has an anti-inflammatory role in atopic dermatitis, an inflammatory disease in which there is an alteration of the cytoplasmic aryl hydrocarbon receptor (AhR) pathway implicated in the regulation of the expression of several genes that influence skin inflammation. In fact, it has been shown that altered AhR signaling is involved in the progression of atopic dermatitis, with impairment of its skin barrier function and increased immune activity [93,94]. RES is able to act as a ligand on the AhR receptor, which, by activating the AhR-OVOL1 pathway, restores the skin barrier [95] and downregulates proinflammatory cytokines to reduce inflammation [96].

Thus, RES can modulate several inflammatory signaling pathways, regulating the expression of numerous pro-inflammatory genes and reducing the expression of pro-inflammatory cytokines involved in chronic inflammation. It has been shown to also reduce inflammatory marker levels, such as C-reactive protein (CRP) levels, which are often elevated in inflammatory conditions [97].

#### 2.1.2. RES and the Cardiovascular System

Another key pharmacological target of RES is cardiovascular diseases. RES is associated with cardioprotective effects, such as promoting endothelial function [98], improving left ventricular systolic and diastolic function and flow-mediated dilation, inhibiting platelet aggregation, and reducing low-density lipoprotein (LDL) levels [99]. These properties may help prevent atherosclerosis, coronary artery disease, and hypertension [98,100]. Furthermore, RES reduces lipid accumulation and inflammation by decreasing the expression of intercellular adhesion molecules (ICAM), vascular cell adhesion molecules (VCAM), and IL-8, which contribute to atherosclerosis development [101]. In a study on smokers [102], RES reduced systemic airway inflammation and CRP levels released by the liver. Despite these positive cardiovascular effects, some studies have reported conflicting results. For instance, in a study on obese schizophrenic patients [78], RES worsened lipid and cholesterol profiles. Additionally, in overweight individuals, RES did not influence high-density lipoprotein (HDL) and apoA-I levels [103]. Moreover, RES supplementation during post-ischemic recovery could represent an important therapeutic strategy for cardiovascular protection [104].

#### 2.1.3. RES and Neuroinflammation

Due to its ability to cross the blood–brain barrier [105,106], RES has been investigated for its potential in neurological diseases. Its anti-inflammatory and antioxidant effects may be useful in treating conditions such as Alzheimer’s disease and stroke. Furthermore, RES modulates the activities of Sirtuin 1 (SIRT1), AMPK, and PGC-1α, which are involved in these diseases. In RES-treated individuals, a reduced accumulation of beta-amyloid in the brain has been observed [107,108], suggesting a potential neuroprotective role in Alzheimer’s disease. Some studies [105,106,109] have shown that RES reduces matrix metalloproteinase (MMP)-9 levels, an enzyme associated with neurodegeneration. Additionally, MMP regulation by RES is exploited in combination with recombinant tissue plasminogen activator (r-tPA) for treating ischemic stroke patients [110]. Lastly, RES is able to suppress neuroinflammation and alleviate airway inflammation due to asthma [67,111].

#### 2.1.4. RES, Diabetes, and Metabolism

RES may also be effective against diabetes [61,112,113]. One study found that RES reduced glycated hemoglobin (HbA1c), cholesterol, and systolic blood pressure levels, improving glycemic control [114]. In animal models, RES increased glucagon-like peptide-1 (GLP-1) regulation [115], reduced insulin resistance and blood glucose levels, and delayed postprandial glucose peaks [116].

Due to its ability to counteract insulin resistance, RES has been proposed as a treatment for non-alcoholic fatty liver disease (NAFLD). Other pharmacological targets of RES against NAFLD include anti-inflammatory effects, SIRT1 activation, and caloric restriction-mimicking properties [117]. Additionally, RES reduced alanine aminotransferase (ALT) and aspartate aminotransferase (AST) levels in NAFLD patients [118].

#### 2.1.5. RES and Gut

In animal models, RES has shown positive effects on gut health [119], improving microbiota composition [120] and increasing jejunal villus height [121]. RES also exhibits antimicrobial properties, counteracting both acute and chronic inflammation [122,123].

Moreover, RES dietary supplementation can have an anti-inflammatory effect by acting on the intestinal barrier function through the regulation of genes related to intestinal tight junctions and adherence junctions [124].

#### 2.1.6. RES, Muscle Strength, and Endurance

RES also enhances muscle strength and increases endurance [125]. RES enhances muscle function by increasing contractile protein accumulation and improving muscle strength and fiber cross-sectional area [126]. After muscle contusions, RES reduces creatine kinase and lactate dehydrogenase levels in the serum, promoting cell regeneration [127].

#### 2.1.7. RES and Cancer

In the last decades, numerous studies have established a close correlation between cancer development and inflammation [128,129,130]. Since the development of anti-cancer effective therapies with a long-lasting response still represents a goal not fully achieved, the prevention of carcinogenesis is a topic of great interest for researchers. For this purpose, natural products appear very promising [131], as they exhibit fewer side effects compared to synthetic drugs [132]. In this respect, RES, due to its strong anti-inflammatory power, could represent an excellent candidate for the prevention and/or treatment of cancer. The first work that gave us information on this topic dates back to 1997, in which Jang and colleagues demonstrated that in a mouse skin cancer model, RES was able to inhibit tumorigenesis [133]. Since then, many works have demonstrated the tumor-inhibiting capacity of RES.

The anti-inflammatory effects of RES are caused by its ability to regulate several pathways, including NF-κB, resulting in attenuation of NF-κB signaling [134,135]. The NF-κB pathway deregulation has been shown to be related to cancer development [136] by playing a role in tumor growth and invasion [124,137,138,139].

Alterations of the Phosphatidylinositol-3-kinase (PI3K)/protein kinase B (AKT) pathway are present in many types of human tumors, regulating both the proliferation and cell survival of tumor cells and the response to anti-tumor treatment [140]. AKT regulates the transcriptional activity of NF-κB by inducing the phosphorylation and subsequent degradation of IκB, its inhibitor [141]. In addition to acting directly on NF-κB, it has been shown that, in tumors, RES also acts on the PI3K/AKT pathway by inhibiting it, thus regulating cellular differentiation, growth, and proliferation of tumor cells [142,143,144]. RES, by down-regulating the PI3K/AKT/NF-κB pathway, inhibits the invasion of glioblastoma-initiating cells in glioblastoma [145], and in human glioma cells U373M, it regulates NF-κB-dependent TNF-α expression by reducing cell invasion [138]. The inhibition of the AKT signaling pathway is also utilized in the administration of RES in combination with rapamycin. Although rapamycin inhibits the mechanistic target of rapamycin (mTOR), it is not effective as monotherapy since tumor cells can activate alternative survival pathways, such as AKT signaling [146]. The combination of RES and rapamycin has been successful in studies on bladder and breast cancer cell lines, lymphangioleiomyomatosis cells, TSC2-deficient cell lines, and TSC2-deficient xenograft tumors in mice [9,146].

Furthermore, RES activates apoptosis through the down-regulation of Bcl2 and Stat3 and promotes the up-regulation of SOCS, acting on the JAK-STAT pathway in squamous cell carcinoma cells [147].

The promising results from in vitro studies are not always confirmed by clinical findings [148]. For instance, in the study by Paller et al. [149], RES was reported to delay recurrence, increasing PSA doubling time by 5.3 months. In contrast, Kjaer et al. [150] found no significant effects of RES on prostate cancer, as it did not alter PSA levels or prostate volume.

In some studies on colorectal cancer cells and liver metastases, RES [151,152] induced apoptosis and inhibited cell proliferation. However, caspase-3 and Ki-67 levels were only slightly altered [75,153], casting doubt on the potential use of RES in colorectal cancer treatment.

Promising results have been obtained in vitro for multiple myeloma, where RES inhibited the NF-κB, AKT, and STAT3 pathways and exhibited cytotoxicity against myeloma cells [154]. However, in a clinical trial conducted by Popat et al. [155], patients experienced renal failure and other side effects. These adverse reactions were not observed in breast cancer patients, where serum RES levels inhibited the *RASSF-1α* gene, suggesting a potential preventive role in breast cancer [156].

### 2.2. Sirtuins: Molecular Targets of RES and PA

Sirtuins are a family of nicotinamide adenine dinucleotide (NAD^+^)-dependent protein deacetylases and/or ADP-ribosyltransferases consisting of seven members (SIRT1–7) [157]. They were identified for the first time in the yeast *Saccharomyces cerevisiae*, where the silent information regulator 2 (Sir2) plays a role in chromatin remodeling, gene expression, and DNA recombination [158]. In mammals, sirtuins are gathering increasing interest due to their crucial roles in several biological processes like energy metabolism, aging, longevity, reproduction, cell proliferation, differentiation and death, or cancer progression [159,160,161]. As a consequence, the development of “sirtuin-activating compounds” (STACs) has been a major goal of the field [162,163].

Sirtuins are widely expressed in tissues, but their localization within the cell is different, spanning from cytoplasm to nucleus or mitochondria, as summarized in Table 1. Furthermore, the large number of molecular targets inside cells makes sirtuin activity critical for most biological functions like proliferation, differentiation, and death (details in Table 1) [119,164,165,166,167,168].

The expression rate of SIRT1, SIRT3, and SIRT6 is induced by caloric restriction and exercise in animal models or humans [176,177,178,179,180,181]. The over-expression of SIRT1 or SIRT6 in mice mimics many of the physiological effects of physical exercise and caloric restriction, with benefits on life/health span. On the contrary, their genetic ablation or inactivation prevents the benefits of calorie restriction and exercise on health and longevity [164,165,182]. The activity of sirtuins is further protective in disease models of cancer, type 2 diabetes, and neurological or cardiovascular diseases [161,183,184].

Sirtuins mediate metabolic benefits in various tissues [165,185]. Their activity has been described in skeletal muscle, liver, pancreas, adipose tissue, and the brain, with effects on glucose homeostasis in the liver, the metabolic switch from carbohydrate to lipid use for energy production in muscle and white adipose tissue (WAT), mitochondrial biogenesis and fatty acid oxidation in the muscle, or the central control of food intake within the arcuate nucleus in the brain, among the others [161,165,182]. The main signaling pathways of sirtuins as “metabolic sensors” have been elsewhere reviewed [92,164,165] and are briefly summarized in Table 2.

SIRT1 is the best-known and most studied sirtuin. Due to its ability to target histone tails with consequences on chromatin architecture, metabolism-related transcriptional factors (e.g., FoxO1, FoxO3, and mTOR), or transcription co-activators (e.g., PGC-1α) (D’Angelo 2021) among the others, it is considered both an epigenetic target and a metabolic sensor [161,164,165].

In this respect, the modulation of SIRT1 and its related pathways is critical for the adaptive responses to environmental factors and lifestyle, which include PA and diet. Hence, following PA and exercise, SIRTs mediate the adaptive response by deacetylating factors involved in the body’s response to exercise-induced stress [164]. Furthermore, SIRT1 increases the expression of antioxidant genes such as GPX, CAT, and SOD1 through the activation of the nuclear factor E2-related factor 3/antioxidant response element (Nrf2/ARE) pathway and can also modulate the NF-κB pathway by inhibiting the transcription of pro-inflammatory genes [2,186].

SIRT1 has been dubbed the “longevity gene” due to its multiple roles in inflammation, cell cycle, differentiation, survival, apoptosis or autophagy, mitochondrial biogenesis, glucose-lipid balance, metabolism, and aging [187,188,189].

Several RES biological effects on the brain, the cardiovascular system, mitochondrial activity, glucose and lipid metabolism, muscle, etc. occur via the sirtuin-mediated pathways, but most studies have been focused on SIRT1 [190], whose expression increases with RES supplementation in a dose-dependent manner [104]. However, the cellular pathways affected by modulating the activity of each sirtuin isoform in different physio-pathological conditions are not fully known and deserve further studies [190]. As suggested by Motti et al. [21], the supplementation of micro and/or macronutrients may be useful not only to prevent infection but also to support immune response during COVID-19, as well as in the post-acute phase. In fact, RES administration, thanks to its ability to activate SIRT1, has been proposed as a potential approach to fight this viral infection [73].

## 3. Resveratrol and Physical Activity: Pre-Clinical and Clinical Trials

RES has gained attention for its potential health benefits, as it increases exercise performance and exhibits anti-oxidation, anti-inflammatory, and anti-cancer effects [191]. Some studies suggest that RES supplementation may enhance exercise performance by increasing muscle strength, endurance, and power output. This may be due to its ability to improve mitochondrial function and reduce oxidative stress [104]. Recently, there has been an awareness of the know-how of RES to modulate physical performance and prevent oxidative stress [192]. RES may help delay muscle fatigue during exercise by improving blood flow and nutrient delivery to muscles, as well as reducing inflammation [193]. Furthermore, the anti-inflammatory properties of RES may aid in muscle recovery after exercise, reducing soreness and promoting faster healing. The effects of RES on the physiology of muscle and bone, exercise performance, oxidative stress, and metabolism in both animal models and humans have been summarized in the next paragraphs.

### 3.1. Pre-Clinical Trials

All the properties of RES mentioned above have convinced researchers to explore the potential benefits this compound may provide to the locomotor system and in conjunction with physical exercise in animal models (Table 3).

Momken et al. [194] demonstrated the prevention of muscle degradation in rats after supplementation with 400 mg/kg/day of RES, noting a significant reduction in muscle fiber atrophy.

Durbin et al. [195] studied the effect of RES in preventing bone loss and skeletal disuse in elderly male rats. The femurs and tibias of the sample were immobilized for 14 days, which accelerated the loss of bone mineral content, reduced trabecular bone volume per unit of total volume, and increased trabecular separation. In the group that received 12.5 mg/kg of RES, there was an improvement in bone demineralization and the loss of bone microarchitecture.

In Alkhouli et al. [196], 4 g/kg of RES was administered in pellet form, along with physical exercise or not, to mice. Physical exercise was performed on a treadmill at 15 m/min for 45 min/day, 5 days/week. The combination of RES and physical exercise suggested a preventive effect on fracture risk associated with Alzheimer’s disease by inhibiting the accumulation of advanced glycation end-products (AGEs) in the bone extracellular matrix.

Dolinsky et al. [197] compared a control diet with a diet containing 4 g/kg of RES administered to male Wistar rats who were then subjected to treadmill running for 12 weeks. RES supplementation improved exercise performance, increased both the concentrative and tetanic strength in the soleus muscle, increased ejection fraction, and reduced left ventricular wall stress. Additionally, there was an increase in cardiac fatty acid oxidation, favorable changes in cardiac gene expression, and signaling pathways that optimized fatty acid utilization.

Hart et al. [198] subjected High Capacity Runner rats to treadmill training for 12 weeks and administered 100 mg/kg of RES orally. RES enhanced aerobic performance and upper limb strength, activated AMPK, SIRT1, and mitochondrial transcription factor A, and improved aerobic performance through the AMPK-SIRT1-PGC-1 pathway activation.

Bennet et al. [199] tested muscle recovery in 32-month-old rats after hindlimb suspension for 14 days. The first group was administered 125 mg/kg/day of RES via oral gavage, and another received a placebo. After the period of inactivity, the rats started walking again, and the comparison data showed that RES induced positive changes in the cross-sectional area of type IIA and IIB muscle fibers and reduced apoptotic signaling in the muscles, although it did not prevent the loss of body or muscle weight after hindlimb suspension.

In Su et al.’s [200] study on mice, different doses of RES (25 and 150 mg/kg) were administered along with treadmill exercise. The high-dose RES group prolonged the time before exhaustion during short-duration downhill running. Additionally, high-dose RES reduced the mRNA expression of TNF-α and enhanced the expression of SIRT1, glucose transporter 4 (GLUT4), AMPK α1, and AMPK α2 in certain muscles.

In a study on muscular dystrophy [201], 100 mg/day of RES or a control aqueous solution was administered for 8 weeks. The group receiving RES showed improvements in rotarod performance, peak tension in situ, central nucleation, and oxidative stress.

Amirazodi et al. [202] studied the interactive effects of swimming high-intensity interval training (HIIT) and RES consumption on SIRT3 and SIRT4, NAD^+^/NADH, AMPK, and SOD2 expression in aged rats. Aged rats were divided into a control group, a HIIT swimming group, a HIIT swimming + RES group, a RES consumption group, and a placebo group. RES was administered for 6 weeks at a dose of 10 mg/kg/day via gastric gavage. NAD^+^/NADH, SOD2, and AMPK were significantly increased in the HIIT + RES group. HIIT increased SIRT3, but RES reduced it. On the other hand, SIRT4 was reduced by HIIT, while RES positively influenced it. The authors concluded that the combination of HIIT and RES could have beneficial effects in counteracting aging and oxidative stress in the hippocampus of aged rats.

Previously, Mehrabi et al. [203] found an increase in SIRT3 in rats combining swimming and RES supplementation compared to those receiving only RES.

Lou et al. [204] administered 50 mg/kg of RES via gastric gavage to rats 1 h after swimming (60 min, 6 days a week). Compared to the control group, a reduction in blood urea nitrogen, CK activity, and malondialdehyde content in skeletal muscle was observed, as well as an increase in total SOD activity in skeletal muscle, regulation of the SIRT1/PGC-1α-nuclear respiratory factor 1 pathway, and improved fatigue status and Ca^2+^-Mg^2+^-ATPase, Na^+^-K^+^-ATPase, succinate dehydrogenase, and citrate synthase activities in skeletal muscle.

In Lin et al.’s [205] study, 15 mg/kg/day of RES was administered to 18-month-old rats, who were required to swim for 1 month. In protein analysis, the PI3K-AKT pathway was slightly increased with physical training and RES treatment, but SIRT1 was significantly activated only with RES treatment in the rats’ hearts. Additionally, the physical training + RES group benefited from both the SIRT1 and PI3K-AKT pathways and blocked FoxO3 accumulation. These results showed the protective power of the combination of RES and PA on the hearts of aged rats.

Wu et al. [125] divided male mice into four groups based on the RES dose administered (0, 25, 50, and 125 mg/kg/day) and evaluated differences after 3 weeks of swimming. Exercise-induced fatigue parameters, including lactate, ammonia, glucose, and CK, were positively modulated by RES supplementation in a dose-dependent manner.

Qin et al. [206] compared the effects of RES in its normal form and in solid lipid nanoparticles on male mice subjected to forced running. RES in nanoparticle form showed a protective effect against oxidative stress induced by chemicals on AST, ALT, and lactate dehydrogenase (LDH) activities and blood urea nitrogen and creatinine levels compared to the normal RES and control groups. Additionally, RES in nanoparticle form could also reduce lipid peroxidation and attenuate oxidative damage induced by exercise.

### 3.2. Clinical Trials

The promising results obtained in animal studies prompted researchers to test RES for physical recovery and performance enhancement in humans as well. Below, we will discuss studies on young and untrained adults. Researchers focused on the potential side effects, trying different RES dosages to explore how this polyphenol could help sedentary individuals begin PA more easily, counteracting post-exercise pain, and examining its antioxidant characteristics.

Voduc et al. [207] showed that RES supplementation (500 mg/day for 1 week and 1000 mg/day for 3 weeks) did not produce significant changes in physical performance, blood parameters, inflammatory components, or liver and kidney functions in healthy sedentary adults. Furthermore, gastrointestinal side effects were observed in the RES group, especially for those who took the higher RES dose.

Alipour Ghazichaki et al. [208] administered 500 mg/day of RES to overweight men for 8 weeks. The sample practiced Pilates three times a week. In the group combining RES and Pilates, there was a significant increase in sestrin 2 (SESN2) and GPX and a decrease in lipocalin-2 (LCN2), the homeostatic model assessment of insulin resistance (HOMA-IR), and malondialdehyde (MDA). These results suggest that the combination of RES and Pilates can counteract oxidative stress and may be effective in preventing metabolic syndrome and cardiovascular diseases in overweight individuals.

Huang et al. [191] evaluated the effect of different doses of RES (500 and 1000 mg/day) on sedentary young adults 7 days before performing plyometric exercise. At the time of 72 h post-exercise, the peak strength and rate of force development in the countermovement jump in the RES groups showed no significant differences compared to baseline but were significantly greater than in the placebo group. RES supplementation had a better recovery effect on peak power, mean relative power, and fatigue index, especially in the group receiving 1000 mg of RES. Additionally, post-exercise pain perception was lower in the RES groups. Muscle damage indices such as creatine kinase and lactate dehydrogenase were lower in the RES group, indicating a potential for faster recovery.

Research on trained individuals mainly focused on how RES could improve physical performance and recovery, counteracting oxidative stress and inflammation caused by sports practice.

Macedo et al. [209] conducted a double-blind placebo-controlled study on Brazilian soldiers. They were given 100 mg/day of RES for 90 days. Analyses were performed before and after a fitness test. The group receiving RES showed no hepatic consequences compared to the placebo group after analyzing AST, ALT, and gamma-glutamyl transpeptidase (GGT) plasma activities. Blood sugar levels increased after exercise with RES supplementation. CK levels increased in the placebo group but not in the RES group. RES supplementation reduced IL-6 and TNF-α levels, confirming its anti-inflammatory effect.

In Tsao et al.’s [210] study, 480 mg/day of RES for 4 days was administered to cyclists. A reduction in IL-6 levels was observed. However, no other significant changes were distinguished in blood tests or physical performance. It is interesting that even with such a short duration of RES treatment, anti-inflammatory effects were evident, although 4 days were not sufficient to attenuate oxidative stress and fatigue from exercise.

Kristoffersen et al. [211] administered 500 mg/day of RES or placebo (400 mg of Calcium) to young untrained adults 3 days before the first test. All subjects underwent a maximal eccentric dorsiflexion protocol (10 × 10) to induce muscle damage and delayed onset muscle soreness. The group receiving RES experienced reduced pain perception, but there were no differences with the placebo group in terms of muscle strength or activity.

Laupheimer et al. [212] studied the effect of 600 mg/day of RES on runners who were given the supplement 7 days before the London Marathon. The RES group showed no differences in inflammatory response (white blood cells and CRP or delayed onset muscle soreness compared to the placebo group). However, the authors noted that the sample size was not statistically significant and that measurements beyond 32 h after the marathon would have been needed, and the RES dose might have been insufficient.

As previously mentioned, physical exercise is essential for health, including in older age groups. Studies on the elderly have assessed how the combination of RES and exercise could be utilized to counteract age-related diseases, especially those affecting skeletal muscles.

Gliemann et al. [213] compared the effect of 250 mg/day of RES in elderly individuals by dividing the sample into two groups: one inactive and one engaged in intensive training for 8 weeks. These were further subdivided based on the RES or placebo supplementation they received. The exercise group with RES supplementation did not show an increase in capillary-to-fiber ratio or muscle vascular endothelial growth factor (VEGF) protein. Tissue inhibitor of metalloproteinase-1 (TIMP-1) protein levels were lower in the exercise + RES group compared to the exercise + placebo group. These data suggest that RES supplementation might limit both baseline and exercise-induced angiogenesis.

Alway et al. [214] administered 500 mg of RES to healthy individuals aged 65 to 80 years after exercise. RES improved mitochondrial density and muscle fatigue resistance indices more than placebo and exercise alone. The combination of exercise with RES resulted in increased peak torque, mean peak torque, and post-training power in the knee extensor muscle. Additionally, the RES group showed improvements in the average fiber area and total myonuclei in the vastus lateralis muscle fibers. However, the combination of aerobic and resistance exercise with RES did not further reduce cardiovascular risk compared to exercise alone. Considering these results, the combination of RES and exercise may be beneficial in counteracting sarcopenia.

Harper et al. [215] assessed how different doses of RES (0, 500, and 1000 mg/day for 12 weeks) administered to elderly individuals with physical functional limitations could affect exercise (walking and resistance training). RES supplementation improved mitochondrial function in skeletal muscle and physical function indices related to mobility in a dose-dependent manner.

Nicolau et al. [216] evaluated the effect of 300 mg/day of RES for 60 days in women aged 60 to 80 years who practiced physical exercise. No significant differences were observed in anthropometric parameters, heart measures, blood analytes, leukogram, and platelet indices, except for a mild anti-inflammatory effect. Furthermore, in the group of women who did not engage in physical exercise but took RES, an increase in both systolic and diastolic blood pressure was observed, although this did not exceed pathological values for their age group.

Below is a summary table (Table 4).

## 4. Conclusions

RES is a natural polyphenol found in foods such as grapes, red wine, peanuts, and berries. It has been widely studied for its potential health benefits and well-being, due to its antioxidant, anti-inflammatory, and cardioprotective properties [217], making it particularly interesting in the management of many chronic diseases, including cardiovascular disease, diabetes, arthritis, and even in several types of tumors. Its antioxidant and anti-inflammatory properties, along with its potential cardiovascular and cognitive benefits, make it an intriguing compound worth considering.

Antioxidants can help to protect cells from damage caused by exercise, such as muscle soreness and inflammation. The amount of exercise needed to increase antioxidant production is not known for sure. However, most experts recommend at least 30 min of moderate-intensity exercise most days of the week. Exercise can also increase the production of free radicals. However, the body’s antioxidant defenses are usually able to handle the increased free radical production from exercise [143].

RES supplementation has garnered growing interest as a potential strategy for improving physical performance and optimizing muscle recovery. Several animal studies have shown promising effects, with improvements in protection against oxidative stress, muscle capacity, and energy metabolism [125,197]. It is important to note that the research on RES is still ongoing, and more studies are needed to confirm its potential health benefits.

The benefits of RES are likely mediated through the activation of sirtuins, particularly SIRT1, which plays a crucial role in improving resistance to oxidative stress and inflammation [104,186].

RES supplementation in animal models has also highlighted an increase in physical endurance, protection against inflammation, and improvement in muscle mass, suggesting potential benefits for the treatment of metabolic and muscular diseases [194,201].

The findings from animal models are particularly significant, as they have helped clarify the effectiveness of RES in a controlled context, revealing potential therapeutic applications in various fields. For example, RES treatment has shown improvements in muscle mass and strength in elderly rats, counteracting muscle atrophy due to inactivity [199]. These data suggest that RES may have applications in muscle rehabilitation and the treatment of degenerative diseases. However, some animal studies have shown contrasting results [198]. This underscores the importance of optimizing dosages and administration methods to maximize benefits.

Human studies suggest that RES intake combined with physical exercise may improve parameters such as endurance, inflammation reduction, and muscle pain management [125,210].

However, some research highlights the lack of significant benefits in terms of performance or recovery, raising questions about the real effectiveness of RES [212,216]. Additionally, concerns regarding side effects, such as increased blood pressure in elderly individuals, suggest that RES use in certain categories of people may require further evaluation [216]. The optimal dosage and duration of RES supplementation for exercise benefits are not yet established. Most studies have used doses ranging from 150 to 500 mg per day for several weeks or months. Such RES doses are high and can never be achieved with a normal diet; such high amounts of RES can be administered only in the form of drugs or dietary supplements. Therefore, without additional supplementation, it is unlikely to achieve the effects obtained [218]. It is questionable that the non-appearance of metabolic benefits is simply due to differences in the dose or duration of resveratrol supplementation. Moreover, the effects of RES on exercise performance can vary depending on factors such as age, fitness level, and genetics. RES is not a substitute for regular exercise. A healthy lifestyle that includes regular PA and a balanced diet is essential for overall health and well-being. RES is generally considered safe, but some people may experience side effects such as nausea, diarrhea, or headache [18].

In conclusion, although RES shows promising potential to enhance the effects of physical exercise, the variability of results and the limited number of clinical studies indicate that further research is needed to clarify the real benefits and potential risks associated with RES use as a supplement for professional athletes and individuals engaging in PA. The variability in individual responses and administration protocols represents a major challenge in drawing definitive and standardized conclusions.

## Figures and Tables

**Figure 1 nutrients-17-00837-f001:**
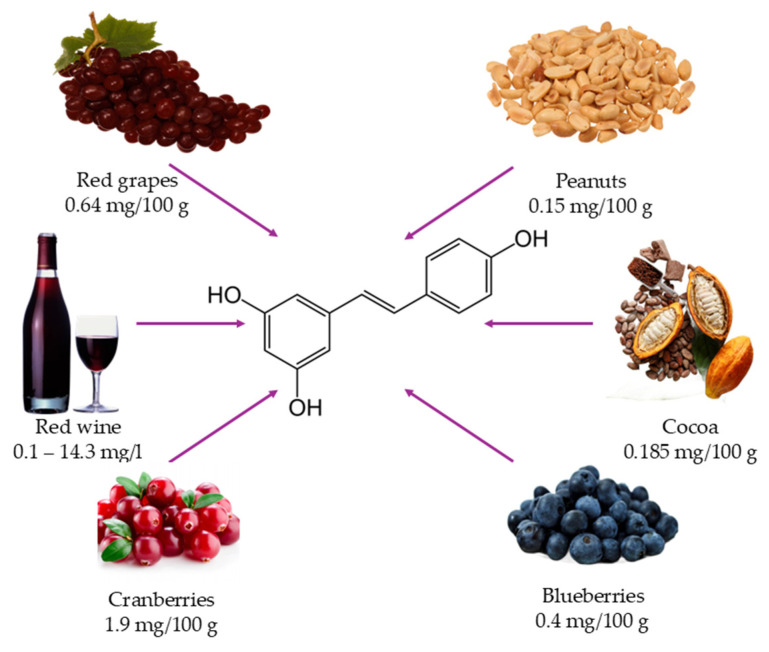
Dietary sources of resveratrol and their respective content (mg/100 g or mg/L) [68].

**Figure 2 nutrients-17-00837-f002:**
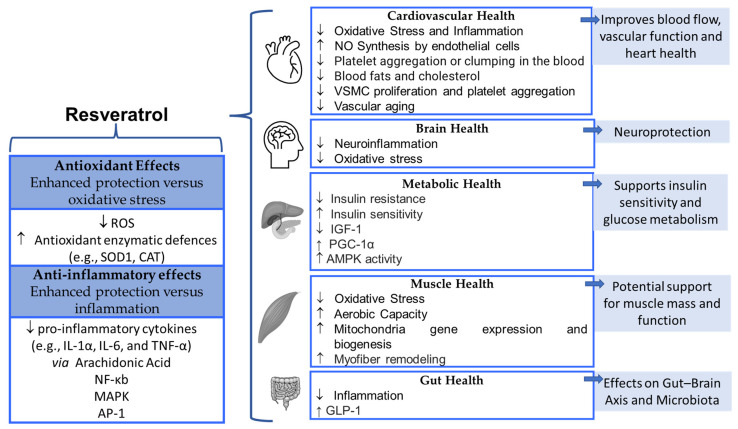
Biological and health effects of Resveratrol. AMPK, AMP-activated protein kinase; AP-1, activator protein-1; CAT, catalase; GLP-1, glucagon-like peptide 1; IGF-1, insulin-like growth factor 1; IL, interleukin; MAPK, mitogen-activated protein kinase; NF-κb, nuclear factor-Κb; NO, nitric oxide; PGC-1α, peroxisome proliferator-activated receptor-γ coactivator 1α; ROS, reactive oxygen species; SOD1, superoxide dismutase 1; TNF-α, tumor necrosis factor α; VSMC, vascular smooth muscle cell; ↑, induce; ↓, reduce.

**Table 1 nutrients-17-00837-t001:** Sirtuins: from cellular localization to biological functions.

Sirtuin	Cellular Localization	Enzymatic Activity	Main Biological Functions
SIRT1 [169]	Nucleus	NAD^+^-dependent protein deacetylase	Modulation of cell cycle, response to DNA damage and oxidative stress, regulation of glucose metabolism, apoptosis and autophagy, gene expression, tumorigenesis, and aging through the modification of histones (i.e., H1, H3, H4, H1K26, H3K9, H3K14, and H4K16) and non-histone proteins like transcriptional factors and coactivators (e.g., p53, NF-κB, FoxOs, and PGC-1α )
SIRT2 [170]	Cytoplasm and nucleus	NAD^+^-dependent protein deacetylase	Control of cell cycle, mitotic S-phase progression, and prevention of precocious mitotic entry (G2/M transition) Control of genomic integrity, microtubule dynamics, cell differentiation, metabolic networks, and autophagy
SIRT3 [171]	Mitochondria (i.e., inner membrane), but also cytosol and nucleus	NAD^+^-dependent protein deacetylase	Regulator of cellular energy metabolism Oxidative stress response Promotes mtDNA transcription following metabolic stress Apoptosis Prevents the formation of cancer cells
SIRT4 [172]	Mitochondria (i.e., matrix), but also cytosol and nucleus	ADP-ribosyltransferase NAD^+^-dependent protein lipoamidase, biotinylase, deacetylase, deacylase, and lipoamidase activity	Regulator of mitochondrial function Modulator of the energy metabolism Promoter of DNA damage repair Inhibitor of inflammatory reactions and apoptosis Suggested role in aging and age-related diseases (e.g., cardiovascular, metabolic, and neurodegenerative diseases and cancer)
SIRT5 [173]	Mitochondria (i.e., matrix) and also cytosol, peroxisomes, and nucleus	NAD^+^-dependent lysine demalonylase, desuccinylase, and deglutarylase that specifically remove malonyl, succinyl, and glutaryl groups on target proteins Weak NAD^+^-dependent protein deacetylase activity	Mitochondrial metabolism and cellular respiration Regulation of glucose metabolism and glycolysis Fatty acid oxidation, amino acid degradation and ROS homeostasis Regulation of blood ammonia levels during prolonged fasting Increased production and biological activity of SOD1 and reduction of oxidative stress
SIRT6 [174]	Nucleus	ADP-ribosyltransferase and NAD^+^-dependent protein deacetylase	DNA repair Maintenance of telomeric chromatin Inflammation Lipid and glucose metabolism
SIRT7 [175]	Nucleus and nucleolus	NAD^+^-dependent protein-lysine deacylase that can act both as a deacetylase or deacylase (desuccinylase, depropionylase, deglutarylase, and dedecanoylase)	Cancer progression Heterochromatin silencing and genomic stability via deacethylation of histones (i.e., H3K18Ac or H3K36) and non-histone proteins (e.g., DDX21, RAN, RRP9, FBL, FKBP5/FKBP51, GABPB1, ATM, CDK9, and POLR1E/PAF53) Transcription of ribosomal RNA (rRNA) at the exit from mitosis DNA damage repair

CDK9, cyclin-dependent kinase 9; DDX21, DExD-box helicase 21; FBL, fibrillarin; FoxOs, forkhead box O; GABPB1, GA binding protein transcription factor subunit beta 1; H, histone; K, lysine; NAD^+^, nicotinamide adenine dinucleotide; NF-κb, nuclear factor-Κb; PGC-1α, peroxisome proliferator-activated receptor γ coactivator 1a; POLR1E, RNA polymerase I subunit E; RRP9, ribosomal RNA processing 9; ROS, reactive oxygen species.

**Table 2 nutrients-17-00837-t002:** Sirtuins mediated metabolic benefits/effects.

Tissue	Molecular Target	Effects
Adipose tissue	PPARγ PGC-1 α Prdm16	↑ Fat mobilization ↓ Fat storage in white adipose tissue ↑ Browning of white adipose tissue Differentiation of brown adipose tissue
Brain (i.e., AgRP and POMC neurons)	FoxOs NF-κb	Modulation of: Food intake Leptin signaling Energy expenditure
Heart	eNOS PPARα	↓ Ischemic tolerance ↑ Hyperthrophy
Liver	CRTC2 FoxOs1 FXR LXRα PGAM-1 PGC-1α SREBP1c	↓ Fatty acid synthesis ↑ Fatty acid use ↑ Gluconeogenesis ↓ Glycolysis ↓ Lipogenesis ↑ Bile acid synthesis Cholesterol homeostasis
Pancreas (i.e., β-cells)	FoxOs UCP2	Modulation of insulin secretion
Skeletal muscle	AMPK LKB1 PGC-1α PPARα	↑ Fatty acid use ↓ Glycolysis

AgRP, agouti-related peptide; AMPK, AMP-activated protein kinase; CRTC2, CREB-regulated transcription coactivator 2; eNOS, endothelial nitric oxide synthase; FoxOs, forkhead box O; FXR, farnesoid X receptor; LKB1, liver kinase B1; LXRα, liver X receptor alpha; NF-κb, nuclear factor-Κb; PGC-1α, peroxisome proliferator-activated receptor γ coactivator 1a; PGAM-1, phosphoglycerate mutase-1; POMC, pro-opiomelanocortin; Prdm16, PR domain containing 16; PPAR, peroxisome proliferator-activated receptor; SREBP1c, sterol regulatory element binding protein 1c; UCP2, uncoupling protein 2; ↑, increase; ↓ decrease.

**Table 3 nutrients-17-00837-t003:** Summary table of animal trials on RES and exercise.

Authors (Year)	Sample Group	Dose of RES	Type of Exercise	Biological Site Investigated	Biomarker Changes
Momken et al. (2011) [194]	Male Wistar rats (N = 20) adults	400 mg/kg/day for 6 weeks	Normal ambulation after hindlimb suspension for 15 days	Blood, skeletal muscle, and bones	Prevention of muscle degradation and bone loss during hindlimb suspension Preservation of muscle mass, mitochondrial function, and insulin sensitivity
Durbin et al. (2014) [195]	Male Fischer 344 × Brown Norway rats (N = 27) 33 months old	12.5 mg/kg bw/day for 21 days	Normal ambulation after hindlimb suspension for 14 days	Blood and bones	Reduction in bone mineral loss and microstructural deterioration in aged rats
Alkhouli et al. (2019) [196]	Male triple-transgenic mice (N = 41) 3 months old	146 mg/kg/day for 4 months	Treadmill running	Bones	RES combined with exercise reduced fracture risk by inhibiting AGEs accumulation in bone
Dolinsky et al. (2012) [197]	Male Wistar rats (N = 50) 8 weeks old	146 mg/kg/day 12 weeks	Treadmill running	Blood, skeletal muscle, and cardiac muscle	Improved exercise performance, muscle strength, and cardiac function Increased fatty acid oxidation and beneficial changes in cardiac gene expression
Hart et al. (2013) [198]	Male rats phenotyped for intrinsic treadmill running capacity (N = 48) 13 months old	100 mg/kg/day for 16 weeks	Treadmill running	Skeletal muscle	Enhanced aerobic performance and upper limb strength Activation of the AMPK-SIRT1-PGC-1α pathway
Bennet et al. (2013) [199]	Male Fisher 344 × Brown Norway rats (N = 36) 32 months old	125 mg/kg/day for 14 days	Normal ambulation after hindlimb suspension for 14 days	Blood and skeletal muscle	Improved muscle fiber cross-sectional area and reduced apoptotic signaling No prevention of muscle or body weight loss after hindlimb suspension
Su et al. (2023) [200]	C57BL/6J mice (N = 24) 6 weeks old	25 and 150 mg/kg/day for 4 weeks	Treadmill running	Skeletal muscle, liver, and adipose tissue	High-dose RES prolonged endurance, reduced TNF-α, and increased SIRT1, GLUT4, and AMPK expression in muscles
Gordon et al. (2014) [201]	Male mdx mice (N = 20) 4–5 weeks old	100 mg/kg/day for 8 weeks	Rotarod performance test	Skeletal muscle	Improved rotarod performance, peak tension, central nucleation, and oxidative stress in mdx mice
Amirazodi et al. (2022) [202]	Male Wistar rats (N = 45) 20 months old	10 mg/kg/day for 6 weeks	Swimming high-intensity interval training	Blood and skeletal muscle	Increased NAD^+^/NADH, SOD2, and AMPK with RES + HIIT RES reduced SIRT3 but increased SIRT4
Mehrabi et al. (2021) [203]	Male Wistar rats (N = 30) 20 months old	10 mg/kg/day for 6 weeks	Swimming high-intensity interval training	Blood and skeletal muscle	Increased SIRT3 in rats combining swimming and RES supplementation
Lou et al. (2023) [204]	Male Sprague-Dawley rats (N = 48) 6–8 weeks old	50 mg/kg/day for 6 weeks	Swimming training	Blood and skeletal muscle	Improved fatigue resistance, reduced oxidative stress, and enhanced mitochondrial function
Lin et al. (2014) [205]	Male Sprague-Dawley rats (N = 24) 18 months old	15 mg/kg/day for 4 weeks	Swimming training	Cardiac muscle	RES + exercise activated SIRT1 and PI3K-AKT pathways, providing cardioprotective effects
Wu et al. (2013) [125]	Male ICR mice (N = 32) 6 weeks old	25, 50, and 125 mg/kg/day for 21 days	Swimming training	Blood, skeletal muscle, cardiac muscle, liver, and kidneys	Enhanced endurance and reduced lactate, ammonia, and CK in a dose-dependent manner
Qin et al. (2020) [206]	Male C57BL/6J mice (N = 24) Adults	10 mg/kg/day (as RSV-SLNs) for 8 weeks	Treadmill running	Blood, skeletal muscle, and liver	Nanoparticle-formulated RES provided superior protection against oxidative stress and exercise-induced damage

AGEs, advanced glycation end-products; AKT, protein kinase B; AMPK, AMP-activated protein kinase; CK, creatine kinase; GLUT4, glucose transporter 4; HIIT, swimming high-intensity interval training; PGC-1α, peroxisome proliferator-activated receptor-γ coactivator 1α; PI3K, phosphatidylinositol-3-kinase; SOD, superoxide dismutase; TNF-α, tumor necrosis factor α.

**Table 4 nutrients-17-00837-t004:** Summary table of human trials on RES and exercise.

References (Year)	Sample Group	Dose of RES	Type of Exercise	Biomarker Changes	Side Effects
Voduc et al. (2014) [207]	Healthy sedentary men (N = 6) and women (N = 6) 42.7 ± 9.4 years old	500 mg/day for 1 week and 1000 mg/day for 3 weeks	Incremental exercise test on an electronically braked cycle ergometer	No significant changes in physical performance, blood parameters, inflammation, or liver and kidney functions	Gastrointestinal with higher dose Small elevations in liver enzymes, triglycerides, and total cholesterol
Alipour Ghazichaki et al. (2023) [208]	Overweight men (N = 40) 42.74 ± 5.70 years old	500 mg/day for 8 weeks	Pilates Training	Increase in SESN2 and GPX, decrease in LCN2, HOMA-IR, and MDA	Not detected
Huang et al. (2021) [191]	Young non-athletic males (N = 36) 21.09 ± 1.33 years old	500 e 1000 mg/day for 7 days before exercise	Plyometric exercise protocol	Improvement in recovery, reduction in post-exercise pain, and lower muscle damage markers (CK, LDH), with better results in the 1000 mg group	Not detected
Macedo et al. (2015) [209]	Military firefighters (N = 60) 19–24 years old	100 mg/day for 90 days	Fitness test: chin-up; abdominal sit-up; speed test: 50 m sprint; aerobic exercise: running for 12 min	Reduction in IL-6 and TNF-α in the RES group, with no effects on CK levels post-exercise	Slightly increased the glucose level
Tsao et al., (2021) [210]	Physically active male students (N = 8) 19.2 ± 0.5 years old	480 mg/day for 4 days	Cycling challenge (Monark Exercise, Varberg, and Sweden)	Reduction in IL-6 levels, but no other significant changes in blood parameters or performance	Not detected
Kristoffersen et al. (2022) [211]	Healthy young men (N = 10) and women (N = 8) 22.8 ± 1.1 years old	500 mg/day for 3 days prior to the first measurement	Maximal dorsiflexors voluntary isometric test	Reduced muscle pain perception in the RES group, with no effects on strength or muscle activity	Not detected
Laupheimer et al. (2014) [212]	Healthy male athletes (N = 7) 40–55 years old	600 mg/day for 7 days before the marathon	London Marathon	No significant differences in inflammatory response or muscle soreness post-marathon	Not detected
Gliemann et al. (2013) [213]	Inactive aged med (N = 27) 65 ± 1 years old	250 mg/day for 8 weeks	High-intensity interval training (cycle ergometer) and full body circuit training (Crossfit)	No significant improvement in muscle angiogenesis or VEGF protein	RES resulted in a lower increase in maximal oxygen uptake
Alway et al. (2017) [214]	Healthy older men (N = 12) and women (N = 18) ≥65 years old	10 mg/kg/day for 6 weeks	Swimming high-intensity interval training	Improvement in mitochondrial density and muscle fatigue resistance	RES had adverse effects on improvements in maximal oxygen uptake, on blood pressure reduction and on the lowering of blood lipids induced by PA
Harper et al. (2021) [215]	Older adults (N = 60) ≥65 years old	500 e 1000 mg/day for 12 weeks	Walking and whole-body resistance exercise training program	Improvement in mitochondrial function and mobility in a dose-dependent manner	Gastrointestinal with higher dose
Nicolau et al. (2022) [216]	Women (N = 43) 60–80 years old	300 mg/day for 60 days	General exercise in a community center	Mild anti-inflammatory effect	RES increased blood pressure in women not doing exercise

CK, creatine kinase; GPX, glutathione peroxidase; HOMA-IR, homeostatic model assessment of insulin resistance; IL-6, interleukin-6; LCN2, lipocalin-2; LDH, lactate dehydrogenase; MDA, malondialdehyde; SESN2, sestrin 2; TNF-α, tumor necrosis factor α; VEGF, vascular endothelial growth factor.

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
