# Peer review of "Resveratrol and Physical Activity: A Successful Combination for the Maintenance of Health and Wellbeing?"

_nutrients, 2025, doi:10.3390/nu17050837_

Round 1

Reviewer 1 Report

Comments and Suggestions for Authors

The manuscript presents a non-systematic review of the health effects of resveratrol with an insight into the mechanisms of action of this compound, concentrating on the effects of resveratrol on the physical exercise. The review is interesting and informative. I have the following personal remark to the results of resveratrol supplementation which are not unequivocal. There is a view represented i.a. by the group of J. Vina (Valencia) that  “exercise is the best antioxidant”. According to this view, limited stress and damage induced by physical exercise is beneficial for the body because it induces mounting endogenous antioxidant defense. Thus, antioxidants should rather not be administered during regular training but may be beneficial in case of extreme exercise, limiting the oxidative stress of high intensity and protecting against the damage it can cause. I wonder whether some of the data presented in the review could be considered from that point of view.

Remarks:

Lines 88-90: „regulate other antioxidant enzymes like glutathione peroxidase (GPX), catalase (CAT), and superoxide dismutase (SOD)”, regulate in which way?

Line 115: “wich is wich” does not seem optimal here

Lines 227-228: “RES reduces creatine kinase and lactate dehydrogenase levels”, I guess in the blood plasma

Table 1: “Activation of SOD1 “, what kind of? Stimulation of biosynthesis or direct activation of the protein?

Please check the completeness of the list of abbreviation. It lacks some acronyms, e.g. r-TPA, MMP, ALT, AGE etc.

Line 315: “ther”

Table 2: ” eNOs“,please correct to “eNOS”

Table 3:Please indicate the tissue/material where the changes were observed (e.g. increase in SOD activity, reduced ammonia concentration).

Sometimes the names start from a capital letter, for no obvious reason.

References should be formatted according to the requirements of the journal (abbreviated journal titles, abbreviations with periods).

Comments on the Quality of English Language

The language is not bad but a native speaker could suggest better phrases in some places and correct the use ofcapital letters

Author Response

Rewiewer 1

The manuscript presents a non-systematic review of the health effects of resveratrol with an insight into the mechanisms of action of this compound, concentrating on the effects of resveratrol on the physical exercise. The review is interesting and informative. I have the following personal remark to the results of resveratrol supplementation which are not unequivocal. There is a view represented i.a. by the group of J. Vina (Valencia) that  “exercise is the best antioxidant”. According to this view, limited stress and damage induced by physical exercise is beneficial for the body because it induces mounting endogenous antioxidant defense. Thus, antioxidants should rather not be administered during regular training but may be beneficial in case of extreme exercise, limiting the oxidative stress of high intensity and protecting against the damage it can cause. I wonder whether some of the data presented in the review could be considered from that point of view.

Response:

Thank you for telling as about this study, it’s an interesting insight. We took the opportunity to cite it in the introduction

Remarks:

 Lines 88-90: „regulate other antioxidant enzymes like glutathione peroxidase (GPX), catalase (CAT), and superoxide dismutase (SOD)”, regulate in which way?

Response: Revised accordingly: “up-regulate the activity of…..”

Line 115: “wich is wich” does not seem optimal here

Response: Corrected accordingly

Lines 227-228: “RES reduces creatine kinase and lactate dehydrogenase levels”, I guess in the blood plasma

Response: Revised accordingly: RES reduces creatine kinase and lactate dehydrogenase levels in the serum

Table 1: “Activation of SOD1 “, what kind of? Stimulation of biosynthesis or direct activation of the protein?

Response: Revised as follows: Increased production and biological activity of SOD1 and reduction of  oxidative stress

Please check the completeness of the list of abbreviation. It lacks some acronyms, e.g. r-TPA, MMP, ALT, AGE etc

Response: We have added more abbreviations and arranged them in alphabetical order

Line 315: “ther”

Response :

Corrected : the

Table 2: ” eNOs“,please correct to “eNOS”

Response:  Corrected accordingly

Table 3: Please indicate the tissue/material where the changes were observed (e.g. increase in SOD activity, reduced ammonia concentration).

Response:

In Table 3  We have added a column with the biological sites examined.

Sometimes the names start from a capital letter, for no obvious reason.

Response:

Corrected accordingly

References should be formatted according to the requirements of the journal (abbreviated journal titles, abbreviations with periods).

Response:

Corrected accordingly

Comments on the Quality of English Language

The language is not bad but a native speaker could suggest better phrases in some places and correct the use of capital letters

We sincerely appreciate the reviewer’s observations. We have refined this manuscript and English language has been revised by English language native speaker  before final approval of their submission

Reviewer 2 Report

Comments and Suggestions for Authors

The presented review is very interesting, was well-designed and written. Although there are many different review articles regarding resveratrol, this paper brings something new to the existing body of scientific literature. Therefore, in my opinion, this review article deserves to be published, but before this, some inaccuracies need to be corrected or explained:

  1. line 95 - one bracket is not necessary
  2. line 99 - RES - the abbreviation should be explained here as it this is the first use of the word in the text (I do not count the abstract)
  3. "Figure 2. Food wich is wich resveratrol is found." - something is wrong here
  4. line 118 - "French paradox" should be italic
  5. line 120 - cis-trans isomerism is rather a chemical property, not physicochemical
  6. lines 122-125 - bioavailability refers to the fraction that has entered the bloodstream, therefore, the level of binding to proteins or lipoproteins has nothing to do with reducing bioavailability. They rather affect the reduction of effectiveness. If for some substances the absorption into the bloodstream is around 70%, but later the biological effectivenes is poor, it means that the bioavailability is high, but the distribution is low. Please correct this part because it was written imprecisely
  7. lines 109-112 - please provide detailed information on the most important dietary sources of resveratrol. Maybe in the form of a Table with appropriate references. Moreover please add data regarding dietary intake of RES, as there are studies indicating that dosages administarted during in vivo studies or clinical trials are far above normal intake with food, and therefore such high dosages will never be achieved within normal food consumption
  8. line 284 - "in the in the" - repetition. Please correct. 
  9. Table 4 - as for many studies reported by the Authors administration of high doses of RES resulted in many side effects, please add a separate column to this Table "side effects"
  10. lines 582-583 - please indicate clearly at this point, that such high doses 150-500 mg can never be achieved with a normal diet, and such high amounts of RES can be administarted only in the form of drugs or dietary supplements. It is crucial to understand that dietary intake is not able to provide such high doses as those used in the studies. Therefore, without additional supplementation, it is unlikely to achieve the effects obtained in the experiments

Author Response

Reviewer 2

The presented review is very interesting, was well-designed and written. Although there are many different review articles regarding resveratrol, this paper brings something new to the existing body of scientific literature. Therefore, in my opinion, this review article deserves to be published, but before this, some inaccuracies need to be corrected or explained:

  1. line 95 - one bracket is not necessary

Corrected accordingly

  1. line 99 - RES - the abbreviation should be explained here as it this is the first use of the word in the text (I do not count the abstract)

Corrected accordingly

  1. "Figure 2. Food wich is wich resveratrol is found." - something is wrong here

Figure legend has been corrected. Biological and health effects of Resveratrol are have been signified in the figure

  1. line 118 - "French paradox" should be italic

Changed accordingly

  1. line 120 - cis-trans isomerism is rather a chemical property, not physicochemical

Corrected accordingly

  1. lines 122-125 - bioavailability refers to the fraction that has entered the bloodstream, therefore, the level of binding to proteins or lipoproteins has nothing to do with reducing bioavailability. They rather affect the reduction of effectiveness. If for some substances the absorption into the bloodstream is around 70%, but later the biological effectivenes is poor, it means that the bioavailability is high, but the distribution is low. Please correct this part because it was written imprecisely

Response: In the revised version of the manuscript the sentence has been changes as follows: “In the bloodstream, RES can exist in three different …which can bind to albumin and lipoproteins, further reducing its effectiveness”

  1. lines 109-112 - please provide detailed information on the most important dietary sources of resveratrol. Maybe in the form of a Table with appropriate references. Moreover please add data regarding dietary intake of RES, as there are studies indicating that dosages administarted during in vivo studies or clinical trials are far above normal intake with food, and therefore such high dosages will never be achieved within normal food consumption

Response: We added the concentrations of the most common foods in figure 1.

Additionally, we have included your observation by citing a study in the conclusions.

  1. line 284 - "in the in the" - repetition. Please correct. 

Corrected accordingly

  1. Table 4 - as for many studies reported by the Authors administration of high doses of RES resulted in many side effects, please add a separate column to this Table "side effects"

Response: We have added a column with the side effects.

  1. lines 582-583 - please indicate clearly at this point, that such high doses 150-500 mg can never be achieved with a normal diet, and such high amounts of RES can be administarted only in the form of drugs or dietary supplements. It is crucial to understand that dietary intake is not able to provide such high doses as those used in the studies. Therefore, without additional supplementation, it is unlikely to achieve the effects obtained in the experiments

Response:Text has been improved accordingly.

Reviewer 3 Report

Comments and Suggestions for Authors

This is an interesting review article with quite adequate novelty. However. some points should be adddressed.

  • In the Abstract, the authors should a bith decrease the introduction/background section (first 3 paragraphs), and they add a "Methods" section describing how the collect the existing data and what database used for this purpose, reporting also the main keywords that they used to collect their data.
  • The Introduction section has too high number of paragrphaps. Some of them should be merged according to their thematic issue that the report.
  • In the Introduction section, more data concerning resveratrol should be added beyond polyphenols.
  • At the end of the Introduction section (before the aim of the study), the authors should report the literature gap that exist and that the present review will cover.
  • A methods section should be added after the introduction section
  • The type of review should be reported in both to describe how the collect the existing data and what database used for this purpose, reporting also the main keywords that they used to collect their data.
  • The authors should add the type of review in the Abstract, the Introduction and the methods section (is it a literature or narrative or scoping or systematic review?).
  • The resolution of Figure 2 should be improved. Moreover, it is quite simple. More information could be added.
  • Flow chart diagrams for pre-clinical and clinical studies should be added describing the process of selecting specific papers to be included in the review.
  • A discussion section should be added in which a statistical analysis of the pre-clinical and clinical study could be applied. A brief description of the molecular mechanisms of resveratrol could be provided in this section. 
  • Most of the data provided in Conclusion section could be removed to Discussion section.

Author Response

x

Reviewer 3

This is an interesting review article with quite adequate novelty. However. some points should be adddressed.

  • In the Abstract, the authors should a bith decrease the introduction/background section (first 3 paragraphs), and they add a "Methods" section describing how the collect the existing data and what database used for this purpose, reporting also the main keywords that they used to collect their data.

Changed accordingly

  • The Introduction section has too high number of paragrphaps. Some of them should be merged according to their thematic issue that the report.

Changed accordingly

  • In the Introduction section, more data concerning resveratrol should be added beyond polyphenols.

We included more details on resveratrol in introduction, but most data on resveratrol have been reported in Par. 2.

  • At the end of the Introduction section (before the aim of the study), the authors should report the literature gap that exist and that the present review will cover.

Revised accordingly.

  • A methods section should be added after the introduction section

Methods have been used in the last part of the introduction

  • The type of review should be reported in both to describe how the collect the existing data and what database used for this purpose, reporting also the main keywords that they used to collect their data.

Response:

The submitted manuscript is a narrative review. It has been indicated in the abstract and introduction

             Moreover, a narrative review is a review method in which the researchers summarize different primary studies from which conclusions may be drawn in a systematic way.

  • The authors should add the type of review in the Abstract, the Introduction and the methods section (is it a literature or narrative or scoping or systematic review?).

Changed accordingly

  • The resolution of Figure 2 should be improved. Moreover, it is quite simple. More information could be added.

We have submitted a new figure 2.

  • Flow chart diagrams for pre-clinical and clinical studies should be added describing the process of selecting specific papers to be included in the review.
  • A discussion section should be added in which a statistical analysis of the pre-clinical and clinical study could be applied. A brief description of the molecular mechanisms of resveratrol could be provided in this section. 
  • Most of the data provided in Conclusion section could be removed to Discussion section.

We thank the reviewer for these valuable suggestions. it could be a starting point for a future work

Nevertheless, we submitted a narrative review, and, in our opinion, the suggested queries are not applicable to this kind of manuscript. Well-done narrative reviews provide a readable, thoughtful, and practical synthesis on a topic. They allow review authors to advance new ideas while describing and interpreting literature in the field. Narrative reviews do not aim to be systematic syntheses that answer a specific and to lead at a discussion

Round 2

Reviewer 3 Report

Comments and Suggestions for Authors

The authors have significantly improved their manuscript.